# SpeechFake: A Large-Scale Multilingual Speech Deepfake Dataset Toward Cutting-Edge Speech Generation Methods

## Abstract

As speech generation technology continues to evolve, the risk of misuse through deepfake audio has become a pressing concern, which underscores the critical need for robust detection methods. However, many existing speech deepfake datasets fall short in terms of size, diversity, and linguistic coverage, limiting the ability of models to generalize effectively to unseen deepfakes. To address these limitations, we present SpeechFake, a large-scale dataset specifically designed for speech deepfake detection. With over 3 million deepfakes totaling more than 3,000 hours of audio, SpeechFake was generated using 40 different speech generation tools, including cutting-edge techniques, and spans 46 languages. This paper provides a detailed overview of the dataset's composition and statistics, emphasizing its scale and diversity. Additionally, we establish baseline results for Speech-Fake and explore how factors such as generation methods, language diversity, and speaker variation influence detection performance. We believe SpeechFake will be a valuable resource for advancing speech deepfake detection research, offering opportunities to explore new detection strategies and improve model robustness across diverse and evolving generation techniques. The dataset will be publicly available soon.

## 1 Introduction

In recent years, speech generation technology has rapidly advanced, with models in text-to-speech (TTS) and voice conversion (VC) systems producing highly natural and high-quality voices (Tan et al., 2021; Triantafyllopoulos et al., 2023; Ju et al., 2024). These systems are increasingly used in virtual assistants, content creation, and language learning, making speech synthesis more accessible and widely adopted. However, as the realism of synthetic voices improves, so does the risk of misuse, especially through speech deepfakes, where synthetic voices are used to impersonate real individuals. Such deepfakes have been employed in fraud (Stupp, 2019), identity theft (Korshunov & Marcel, 2018), and misinformation (Chesney & Citron, 2019), highlighting the significant harm they can cause. Therefore, the growing quality and availability of speech generation systems make the need for robust detection methods more urgent than ever.

A key challenge in developing effective deepfake detection methods is the issue of generalization. Detection models often suffer from substantial performance degradation when confronted with unseen deepfakes (Yamagishi et al., 2021; Müller et al., 2022), which underscores the importance of creating comprehensive datasets to support the development of robust detection systems. However, current datasets for this task come with several limitations. Many publicly available datasets are relatively small, and the generation techniques they include are often outdated or limited, making it challenging for models to detect more advanced deepfake technologies. Moreover, most datasets primarily focus on English or Chinese, offering limited representation of other languages. This lack of linguistic diversity makes it difficult for models to generalize effectively to deepfakes in languages beyond English and Chinese.

To address these limitations, we propose SpeechFake, a large-scale dataset designed to significantly improve both the scale and diversity of data available for speech deepfake detection. The dataset contains over 3 million speech deepfakes, amounting to more than 3,000 hours. These deepfakes

are generated using 30 publicly available speech generation tools and 10 commercial APIs, incorporating cutting-edge techniques capable of producing highly realistic synthetic speech. To support multilingual detection and balance language distribution, SpeechFake is divided into two parts: the Bilingual Dataset (BD), focused on English and Chinese, and the Multilingual Dataset (MD), which spans 46 languages, broadening research opportunities in multilingual environments. Furthermore, unlike most existing datasets that offer only binary labels (real/fake), SpeechFake provides rich metadata, including generation methods, speaker identity, language, and text transcriptions, which facilitates deeper research into the factors that influence deepfake detection and enables other potential use cases.

In addition, we conduct a comprehensive set of experiments to establish a baseline for SpeechFake and examine key factors influencing deepfake detection performance. First, we evaluate overall performance across multiple datasets to assess model generalization to both seen and unseen data (Section 4.2). Next, we analyze cross-generator performance to examine how different speech generation methods affect detection accuracy (Section 4.3). We also investigate cross-lingual performance, exploring how models trained on specific languages perform when exposed to deepfakes in other languages (Section 4.4). Finally, we assess cross-speaker performance to determine the impact of speaker variability on detection robustness (Section 4.5). These experiments establish a strong baseline for SpeechFake and provide valuable insights into the strengths and weaknesses of current detection systems, highlighting areas for future improvement.

We summarize our contributions as follows:

- We introduce SpeechFake, a large-scale and diverse dataset for speech deepfake detection that encompasses a wide range of speech generation methods, integrates cutting-edge techniques, and supports multiple languages.

- We conduct extensive experiments, establishing a baseline for the dataset and analyzing the impact of factors such as generation methods, language diversity, and speaker variation on detection performance.

- SpeechFake provides a valuable resource for developing robust deepfake detection models, demonstrating superior performance on existing datasets. It also supports future research in improving model generalization and advancing detection strategies for emerging techniques.

Table 1: Basic statistics of SpeechFake and its comparison with existing speech deepfake datasets. #utt, #spk, #gen represent number of utterances, speakers and generators, respectively. "-" indicates that the dataset does not specify the number of speakers or generators included. We have clarified this in the table caption for the revised version of the paper.

| Dataset | Year | Deepfake Statistics | | | Languages | Access |
|---------|------|------|------|------|-----------|--------|
| | | #utt | #spk | #gen | | |
| ASVspoof2015 (Wu et al., 2014) | 2015 | 246,500 | 106 | 10 | English | Public |
| FakeOrReal (Reimao & Tzerpos, 2019) | 2019 | 87,285 | 33 | 7 | English | Public |
| ASVspoof2019-LA (Nautsch et al., 2021) | 2019 | 130,378 | 107 | 19 | English | Public |
| WaveFake (Frank & Schönherr, 2021) | 2021 | 117,985 | 2 | 6 | English, Japanese | Public |
| ASVspoof2021-LA (Yamagishi et al., 2021) | 2021 | 148,148 | 67 | 13 | English | Public |
| ASVspoof2021-DF (Yamagishi et al., 2021) | 2021 | 572,616 | 93 | 100+ | English | Public |
| ADD2022 (Yi et al., 2022) | 2022 | 389,419 | - | - | Chinese | Public |
| CFAD (Ma et al., 2024) | 2022 | 231,600 | 279 | 12 | Chinese | Public |
| In-the-Wild (Müller et al., 2022) | 2022 | 11,816 | 58 | - | English | Public |
| ADD2023 (Yi et al., 2024) | 2023 | 273,847 | - | - | Chinese | Public |
| HABLA (Tamayo Flórez et al., 2023) | 2023 | 58,000 | 162 | 6 | Spanish | Public |
| MLAAD (Müller et al., 2024) | 2024 | 82,000 | - | 26 | 23 Languages | Public |
| CD-ADD (Li et al., 2024c) | 2024 | 117,720 | - | 5 | Chinese | Public |
| ASVspoof5 (Wang et al., 2024) | 2024 | 1,211,186 | 1,922 | 32 | English | Restricted |
| VoiceWukong (Yan et al., 2024) | 2024 | 413,400 | - | 34 | English, Chinese | Restricted |
| DFADD (Du et al., 2024a) | 2024 | 163,500 | 109 | 5 | English | Public |
| CVoiceFake (Li et al., 2024a) | 2024 | 1,254,893 | - | 6 | 5 Languages | Public |
| SpeechFake-BD | 2024 | 2,003,016 | 541 | 40 | English, Chinese | Public |
| SpeechFake-MD | 2024 | 1,335,492 | 179 | 6 | 46 Languages | Public |

## 2 RELATED WORK

**Speech Generation** Speech generation, or speech synthesis, can be broadly divided into two main tasks: Text-to-Speech (TTS) and Voice Conversion (VC). TTS systems synthesize human-like, natural-sounding voices from written text. Neural network-based TTS has evolved from CNN/RNN-based models (Oord, 2016; Wang et al., 2017; Shen et al., 2018) to Transformer-based architectures (Li et al., 2019; Ren et al., 2021a), progressing from autoregressive generative models to more advanced frameworks like VAE, GAN, flow, and diffusion models (Prenger et al., 2019; Kong et al., 2020; Kim et al., 2021; Liu et al., 2022). Additionally, TTS has transitioned from using cascaded acoustic models and vocoders (Oord, 2016; Shen et al., 2018; Kong et al., 2020) to fully end-to-end architectures (Ren et al., 2021a; Kim et al., 2021). With the success of large language models (LLMs), recent TTS systems also incorporate LLMs for text-to-token generation (Betker, 2023; Du et al., 2024b; Guo et al., 2024).

While traditional TTS systems were limited to generating speech in a specific voice, newer approaches have enabled multi-speaker generation by incorporating speaker embeddings (Kim et al., 2021; Betker, 2023). Additionally, research has advanced to few-shot and zero-shot TTS, often referred to as voice clone, which allows the generation of speech from text using a voice not seen during training, based on just a small sample of the target voice (Arik et al., 2018; Casanova et al., 2022; Wang et al., 2023; Qin et al., 2023).

In contrast, voice conversion modifies an existing speech sample to match a target speaker's voice while preserving the original content. Unlike TTS, which generates speech from text, VC alters speaker characteristics in audio. Both, however, often share similar neural architectures, like sequence-to-sequence models with attention. VC has evolved from traditional parallel data and statistical methods (Godoy et al., 2011) to more flexible non-parallel approaches, improving conversion quality and adaptability (Kaneko & Kameoka, 2018; Li et al., 2021).

Meanwhile, neural vocoders play a critical role in many speech generation systems, directly generating speech from acoustic features like mel-spectrograms (Kong et al., 2020; gil Lee et al., 2023). Recent research shows that vocoded speech plays a significant role in detecting deepfakes (Frank & Schönherr, 2021; Wang & Yamagishi, 2023; 2024). Therefore, we also include vocoded speech as a part of our dataset.

As shown in Figure 1, we classify speech generation methods into three categories based on the input modality at the inference stage: TTS, VC (Voice Clone or Voice Conversion), and NV (Neural Vocoder). In this classification, TTS refers to systems that generate speech from text (with optional speaker IDs to specify voices). VC focuses on generating speech with a desired speaker identity, whether the content comes from text or another speech sample. Finally, NV generates speech from acoustic features without altering the original speaker's identity.

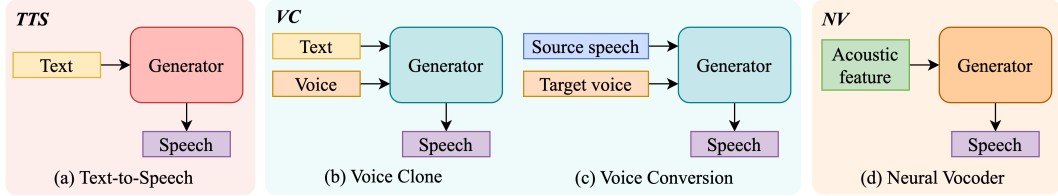

Figure 1: Classification of speech generation methods based on input modality during inference. (a) TTS: Generate speech from text input. (b)(c) VC: generate speech from text or speech based on target voice. (d) NV: Generate speech from acoustic feature.

**Speech Deepfake Datasets** One of the most widely adopted benchmark datasets for speech deepfake detection comes from the ASVspoof challenge (Wu et al., 2014; Nautsch et al., 2021; Yamagishi et al., 2021; Wang et al., 2024). Earlier editions (Wu et al., 2014; Nautsch et al., 2021) primarily focused on spoofing attacks targeting automatic speaker verification (ASV) systems, while more recent editions have expanded to include a wider variety of speech deepfakes that are not constrained by speaker identity (Yamagishi et al., 2021; Wang et al., 2024). Similarly, the Audio Deepfake

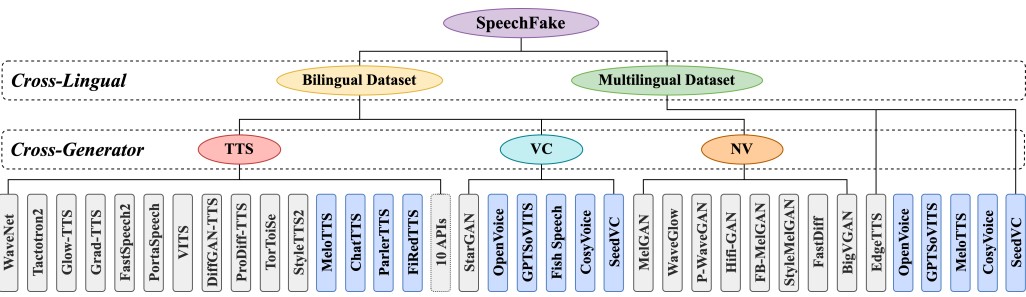

Figure 2: Overview of the SpeechFake dataset. The dataset is divided into two parts: the Bilingual Dataset and the Multilingual Dataset. The Bilingual Dataset is further categorized into three generation methods: TTS, VC, and NV. Methods highlighted in blue represent the latest speech generation methods.

Detection challenge (ADD) has released two editions of datasets with restricted access (Yi et al., 2022; 2024), covering a range of tasks such as deepfake detection, manipulation region localization, and deepfake algorithm recognition. Other notable datasets include FoR (Reimao & Tzerpos, 2019), which collects fake speech from open-source and commercial tools (e.g., Microsoft Azure TTS), and WaveFake (Frank & Schönherr, 2021), which uses neural vocoders to generate synthetic speech. The In-the-Wild dataset (Müller et al., 2022) gathers celebrity synthetic speech samples from the Internet, offering a more naturalistic setting for deepfake detection. For languages beyond English and Chinese, datasets such as HABLA (Tamayo Flórez et al., 2023), MLADD (Müller et al., 2024), and CVoiceFake (Li et al., 2024a) offer valuable resources. HABLA focuses on Spanish, MLADD includes 23 languages, and CVoiceFake spans 5 languages. More recent datasets have focused on latest speech synthesis methods. CD-ADD (Li et al., 2024c) targets zero-shot TTS systems, while DFADD (Du et al., 2024a) emphasizes diffusion-based TTS. The dataset most similar to ours is VoiceWukong (Yan et al., 2024), which contains 6,800 English and 3,800 Chinese deepfake samples generated using 34 different synthesis methods, along with 38 variants such as noise injection and volume control for more robust evaluation.

Although existing datasets have significantly contributed to speech deepfake detection research, many are still limited in scale, linguistic diversity, or the inclusion of advanced generation methods. These limitations hinder the ability to train models that generalize well to modern, multilingual deepfakes. To address these gaps, SpeechFake offers a large-scale, multilingual dataset with cutting-edge generation techniques and rich metadata, enabling more robust and generalizable detection research.

## 3 DATASET COLLECTION AND STATISTICS

### 3.1 DATA COLLECTION

The data collection consists of two parts: real speech, sourced from existing datasets, and fake speech, generated using open-source speech generation methods or commercial APIs. Since most speech generation methods primarily support English or Chinese, we split our dataset into two parts to balance the samples for each language: the Bilingual Dataset, which includes English and Chinese, and the Multilingual Dataset, which covers data from 46 languages. The basic composition of our dataset is illustrated in Figure 2.

**Bilingual Dataset (BD)** The bilingual dataset include English and Chinese speech data. The real data is sampled from four datasets, LibriTTS (Zen et al., 2019) and VCTK (Veaux et al., 2013) for English, AISHELL1 (Bu et al., 2017) and AISHELL3 (Shi et al., 2020) for Chinese. The fake speech data is generated using 30 open-source speech generation methods and 10 commercial APIs, as illustrated in Figure 2 and detailed in Table 7 in the Appendix. The open-source models span a variety of architectures, including GAN-based models (Kumar et al., 2019; Kong et al., 2020), Diffusion models (Liu et al., 2022; Huang et al., 2022b), Sequence-to-Sequence models (Oord,

2016; Ren et al., 2021a), and Flow or VAE models (Prenger et al., 2019; Kim et al., 2021). Besides, we include a collection of latest speech generation methods (highlighted in blue in Figure 2), all of which were released in the past year and represent the cutting-edge in speech synthesis technology.

**Multilingual Dataset (MD)**    The Multilingual Dataset (MD) spans 46 languages in total, including 9 primary languages: English (en), Chinese (zh), Spanish (es), French (fr), Hindi (hi), Japanese (ja), Korean (ko), Persian (fa), and Italian (it), along with 37 additional languages. To create a multilingual setting, we sample real speech data from the CommonVoice dataset (Ardila et al., 2019). The fake data is generated using 6 multilingual speech generation tools, as shown in Figure 2, with EdgeTTS[1] supporting the widest range of languages, while the other methods cover a subset of them.

Before the generation process, we prepare the corresponding text or audio input required by each generator. This input is sampled from the real dataset we collected, including text transcriptions for TTS systems and audio samples for VC and NV systems. The input data is preprocessed as follows:

- **Text Preprocessing**: Text inputs for TTS systems are cleaned by removing special characters, punctuation, and extra spaces. For each language, we ensure that the text maintains an appropriate word or character count (e.g., 5–30 words for English) and is selected to cover a diverse range of phonemes comprehensively. The text is then tokenized and formatted according to the specific requirements of each TTS model, with adjustments for sentence length or phonetic transcription where needed.
- **Audio Preprocessing**: Audio samples for VC and NV systems are resampled to match the generator's required sampling rate and converted to the appropriate formats, such as mel-spectrograms for neural vocoders or raw waveforms for voice conversion models. Silence at the beginning and end of audio clips is trimmed to avoid introducing artifacts during generation.

During the generation process,

- **For TTS systems**: The prepared text is used to generate speech for each method. If the method supports multiple voices, the text is evenly split among the available voices. An exception is TTS_Tortoise, for which additional data is generated to support the cross-speaker experiment.
- **For VC systems**: Reference voices are sampled from the real datasets, while the content comes from the selected text or the corresponding speech recordings. The text is generally split equally among the reference voices. For methods supporting style transfer (e.g., CosyVoice, OpenVoice), we include additional data to reflect the transformed styles.
- **For NV systems**: The generated speech is based on the original input audio selected from the real datasets.

Once the speech is generated, it undergoes the following post-processing steps:

- **Quality Filtering**: We apply voice activity detection (VAD) to filter out speech segments with less than 0.5 seconds of active speech. Additionally, generated speech with noticeable distortions, excessive noise, or unnatural artifacts is discarded.
- **Format Standardization**: The remaining audio clips are standardized to a 16kHz sampling rate, converted to mono, and saved in WAV format to ensure consistency across all samples in the dataset.

## 3.2 DATASET STATISTICS

**Basic Statistics**    We present the basic statistics of SpeechFake and other speech deepfake datasets in Table 1. SpeechFake contains over 3 million speech deepfakes, totaling more than 3,000 hours. The dataset partitioning for experiments is outlined in Table 2. To address the imbalance between the substantial amount of fake data and the comparatively limited real data, and to optimize training efficiency, we utilized approximately half of the fake data for the train, dev, and test sets (split 6:1:3), reserving the remaining portion for future experiments.

---

[1]https://github.com/rany2/edge-tts.git

Table 2: Partition of SpeechFake dataset. Real and fake data are divided into train, dev, test, and optional hidden sets.

| Set | Real Data | | | | Fake Data | | | | |
|---|---|---|---|---|---|---|---|---|---|
| | train | dev | test | total | train | dev | test | hidden | total |
| BD | 75,708 | 12,618 | 37,854 | 126,180 | 633,354 | 105,544 | 315,222 | 898,896 | 1,953,016 |
| BD-UT | - | - | 37,854 | 37,854 | - | - | 50,000 | - | 50,000 |
| BD-EN | 38,400 | 6,400 | 19,200 | 64,000 | 389,866 | 64,970 | 193,461 | 480,585 | 1,128,882 |
| BD-CN | 37,308 | 6,218 | 18,654 | 62,180 | 243,488 | 40,575 | 121,760 | 418,311 | 824,134 |
| BD-TTS | 75,708 | 12,618 | 37,854 | 126,180 | 280,622 | 46,764 | 138,834 | 547,887 | 1,014,107 |
| BD-VC | 75,708 | 12,618 | 37,854 | 126,180 | 192,210 | 32,031 | 96,115 | 214,849 | 535,205 |
| BD-NV | 75,708 | 12,618 | 37,854 | 126,180 | 160,522 | 26,749 | 80,273 | 136,160 | 403,704 |
| MD | 60,000 | 10,000 | 152,757 | 222,757 | 208,126 | 34,690 | 726,136 | 366,540 | 1,335,492 |

The Bilingual Dataset (BD) includes train, dev, and test sets generated using 30 open-source speech generation methods. To assess model generalization to unseen methods, we also created a separate unseen test set (BD-UT) using 10 commercial APIs. The BD is further divided into two language partitions, BD-EN for English and BD-CN for Chinese, as well as three generator-based subsets: BD-TTS, BD-VC, and BD-NV, representing text-to-speech (TTS), voice conversion (VC), and neural vocoder (NV) types, respectively.

The Multilingual Dataset (MD) uses the same generation methods across its train, dev, and test sets but differs in language coverage. The train and dev sets include only English and Chinese data. The test set is divided into 10 subsets: 9 subsets for each of the primary languages and 1 combined subset for the remaining 37 languages. For the latter, around 5,000 clips were selected per language, with the remaining clips hidden for future research.

**Detailed Statistics** We also analyzed the distribution of various attributes within the dataset. Figure 3 illustrates the distribution of the different generation methods used in creating SpeechFake. TTS methods account for the majority, while VC and NV methods represent a smaller portion. On average, each method generates around 60,000 utterances, though some methods produce more to account for a wider range of voices.

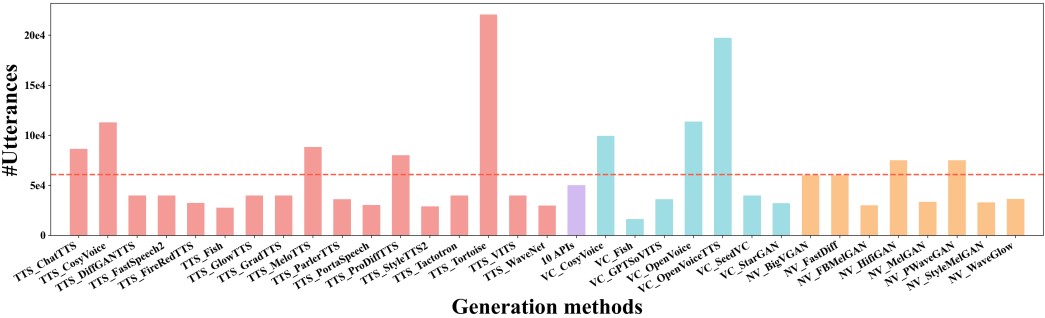

Figure 3: Distribution of generation methods in SpeechFake. The red line represents the average number of utterances across all generation methods.

For the gender distribution of voices, we ensure a balance between female and male speakers. For the languages in the Multilingual Dataset, the 9 major languages account for half of the dataset, with English and Chinese being the most prominent, while the remaining 37 languages make up the other half. As for the audio duration, most clips fall between 2.0 and 20.0 seconds, with some shorter clips (0-2 seconds) and longer ones (>20 seconds), adding variability in length.

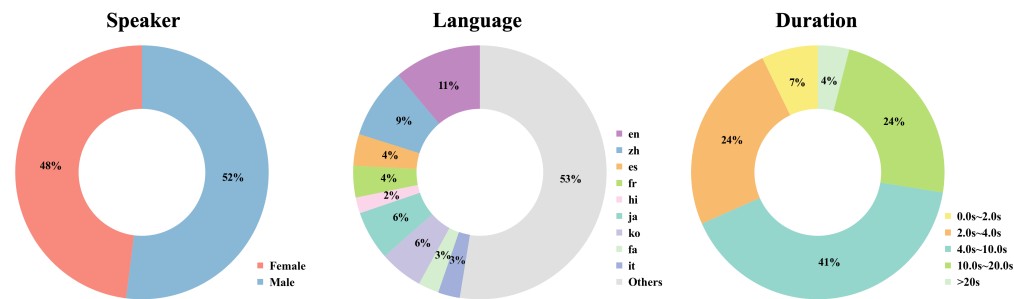

Figure 4: Distribution of speaker gender, language, and duration in SpeechFake. Speaker and duration statistics are based on all data, while language distribution is specific to the Multilingual Dataset.

## 4 EXPERIMENTS AND ANALYSIS

### 4.1 EXPERIMENTAL SETTINGS

To evaluate deepfake detection performance, we use two state-of-the-art models: AASIST (Jung et al., 2022) and W2V+AASIST (Tak et al., 2022). AASIST employs a heterogeneous stacking graph attention network with a novel attention mechanism to capture spoofing artifacts across both temporal and spectral domains. W2V+AASIST integrates Wav2Vec2.0 XLSR (Babu et al., 2021) as a frontend feature extractor with AASIST serving as the backend classifier. The training details for each model are provided in Table 9 in the Appendix. For evaluation, we use the Equal Error Rate (EER) as the metric, following previous work (Yamagishi et al., 2021; Du et al., 2024a).

### 4.2 OVERALL PERFORMANCE

We first establish baseline results to demonstrate the overall performance on the Bilingual Dataset (BD). For training, we include the ASVspoof2019-LA training set (ASV19), which is a widely used benchmark in speech deepfake detection research, alongside three partitions of the BD training set (BD, BD-EN, BD-CN). The evaluation is conducted on multiple test sets: the BD testing sets (BD, BD-EN, BD-CN), and some additional commonly used testing sets in the field: ASVspoof2019-LA eval set (ASV19), In-the-Wild (ITW), and FakeOrReal (FOR). For the FakeOrReal test set, we constructed the test subset by randomly selecting 10,000 utterances due to the relatively small size of the original dataset. Additional results on other test datasets are presented in Table 10 in the appendix.

Table 3: Performance evaluation (EER%) of different models trained on ASVspoof2019 (ASV19) or Bilingual Dataset (BD) across multiple test sets, including both subsets of SpeechFake and other publicly available benchmarks.

| Training Dataset | Model | Testing Dataset (SpeechFake) | | | Testing Dataset (Others) | | |
|---|---|---|---|---|---|---|---|
| | | BD | BD-EN | BD-CN | ASV19 | ITW | FOR |
| ASV19 | AASIST | 39.36 | 41.05 | 39.07 | 1.88 | 45.27 | 36.08 |
| BD | | 3.48 | 3.98 | 2.68 | 23.62 | 7.53 | 23.35 |
| BD-EN | | 9.02 | 6.17 | 12.00 | 30.65 | 6.96 | 28.99 |
| BD-CN | | 16.58 | 24.59 | 5.43 | 16.56 | 8.54 | 25.48 |
| ASV19 | W2V+AASIST | 23.78 | 20.15 | 24.93 | 0.89 | 10.07 | 6.18 |
| BD | | 3.54 | 3.55 | 2.83 | 2.91 | 2.01 | 6.00 |
| BD-EN | | 8.65 | 4.58 | 10.44 | 5.28 | 2.62 | 8.33 |
| BD-CN | | 8.99 | 11.40 | 4.51 | 0.99 | 3.34 | 4.88 |

From Table 3, we observe that when models are trained on ASV19, they perform well on its own evaluation set but experience significant performance degradation on other test sets, particularly on BD, where most of the generation methods are unseen during training. In contrast, training on BD leads to significant accuracy improvements. While training on the English (BD-EN) or Chinese (BD-

CN) subsets yields good performance on their respective test sets, it results in poorer performance on the complementary sets. This may be attributed to the differences in the generation methods or languages included in each partition. Using the full BD training set delivers the best overall results, enhancing accuracy across all BD test subsets compared to training on a single language subset.

When testing on other external datasets, models trained on SpeechFake also demonstrate strong performance. While the EER results on the ASV19 test set do not surpass those of models trained specifically on the ASV19 training set (likely due to the relatively similar distribution between the ASV19 train and test sets), the performance on the ITW and FOR test sets significantly exceeds that of the ASV19-trained models. This indicates that the diversity of the SpeechFake dataset enhances detection accuracy, not only on our own test sets but also on other unseen datasets.

## 4.3 CROSS-GENERATOR PERFORMANCE

To evaluate the impact of generators on detection performance, we conduct cross-evaluations using three categories of generators: TTS, VC, and NV. The results are presented in Table 4.

Table 4: Cross-evaluation performance (EER%) of different generator types as training sets across various testing sets.

| Training Dataset | Model | Testing Dataset | | | | |
|---|---|---|---|---|---|---|
| | | BD-TTS | BD-VC | BD-NV | BD | BD-UT |
| BD-TTS | | 0.44 | 16.85 | 25.66 | 14.26 | 0.53 |
| BD-VC | AASIST | 18.71 | 2.18 | 35.31 | 20.90 | 14.34 |
| BD-NV | | 23.44 | 41.63 | 9.53 | 26.30 | 26.87 |
| BD-TTS | | 1.01 | 9.78 | 14.34 | 8.08 | 0.20 |
| BD-VC | W2V+AASIST | 5.81 | 3.82 | 18.26 | 8.81 | 9.35 |
| BD-NV | | 9.34 | 17.38 | 7.77 | 11.33 | 23.79 |

For each training set, the best detection performance is consistently observed on its corresponding testing set (e.g., model trained on TTS data performs best on the TTS test set), but performance degrades significantly when tested on other generator types. This highlights the challenge of generalizing across unseen generation methods. In terms of overall performance, models trained on TTS data consistently yield the best results on the full BD test set, followed by VC, with NV-trained models performing the worst. This can be attributed to the variety of deepfakes in the TTS subset, which includes cutting-edge techniques that produce highly realistic synthetic speech. On the other hand, NV-based systems likely underperform because the methods used are often outdated and generate lower-quality deepfakes, making it harder for models trained on NV data to detect more advanced techniques. When testing on the unseen commercial TTS API test set (BD-UT), the performance trends remain consistent: TTS-trained models outperform VC and NV. When testing on the unseen commercial TTS API set (BD-UT), TTS-trained models consistently outperform those trained on VC and NV. This highlights that exposure to modern TTS data improves the model's ability to detect high-quality, natural-sounding deepfakes.

In summary, unseen generation methods present a significant challenge for generalization in deepfake detection. Although training on similar generation types can somewhat improve detection performance, substantial differences between generation methods still result in considerable performance degradation.

## 4.4 CROSS-LINGUAL PERFORMANCE

In Section 4.2, the English and Chinese subsets produce distinct results on their respective test sets. This difference can be attributed to variations in both the generation methods and the languages. To further explore the impact of language on deepfake detection, we conducted experiments using the multilingual dataset, where all generation methods are seen during training, but some languages remain unseen.

From Table 5, we observe that both models perform well on the English (en) and Chinese (zh) test sets, with minimal error rates, as expected since these are the seen languages in the training set. For unseen languages, AASIST shows some degradation (e.g., EER = 9.57% for Hindi and 4.90% for French), revealing that language content does influence detection performance, even when

Table 5: Performance evaluation (EER%) on test sets across different languages. The models were trained on English and Chinese.

| Model | Testing Dataset | | | | | | | | | |
|---|---|---|---|---|---|---|---|---|---|---|
| | en | zh | es | fr | hi | ja | ko | fa | it | others |
| AASIST | 0.20 | 0.55 | 1.76 | 4.90 | 9.57 | 1.30 | 0.28 | 2.26 | 1.30 | 2.91 |
| W2V+AASIST | 0.04 | 0.19 | 0.02 | 0.36 | 0.61 | 0.18 | 0.00 | 0.14 | 0.02 | 0.15 |

the generation methods are seen during training. On the other hand, W2V+AASIST demonstrates consistently strong performance across all test sets, including the unseen languages. This can likely be attributed to the multilingual pretraining of the Wav2Vec 2.0 XLSR model (Babu et al., 2021), which helps it generalize more effectively to new languages and mitigate the impact of unseen language content.

These results suggest that language content does impact detection performance, even when the generation methods have been seen during training. However, when the model has prior exposure to the language, such as through multilingual pretraining, this impact can be further mitigated.

## 4.5 CROSS-SPEAKER PERFORMANCE

Some TTS systems are limited to generating specific voices, making it possible for us to detect deepfakes by merely recognizing the speaker's voice rather than learning the distinct audio characteristics that differentiate real and fake speech. This raises the question: can a model learn to detect deepfakes based on their inherent characteristics, or does it simply overfit to the speaker identity?

To explore this, we created a small dataset selected from the Bilingual Dataset (BD). To minimize the influence of different generation methods, we exclusively used TorToiSe (Betker, 2023), a TTS system that supports multi-speaker speech generation. The training dataset is a subset of the BD train set, consisting of 100 real speakers and 10 fake speakers, with a total of 34,305 utterances. As detailed in Table 6, we designed five different test trials, varying the combinations of seen and unseen speakers to assess the model's ability to generalize across speakers. For evaluation, we trained an AASIST model for 50 epochs on this training set.

Table 6: Statistics and EER(%) results of cross-speaker testing trials. #utt, #spk represent number of utterances and speakers, respectively. The numbers in parentheses represent the distribution of speakers (seen, unseen) in the training set.

| No. | Test Setting | Real | | Fake | | EER(%) |
|---|---|---|---|---|---|---|
| | | #utt | #spk | #utt | #spk | |
| 1 | Seen Real & Fake Speakers | 6,599 | 100 (100, 0) | 13,871 | 10 (10, 0) | 0.06 ± 0.01 |
| 2 | Unseen Real & Fake Speakers | 5,557 | 100 (0, 100) | 12,377 | 10 (0, 10) | 0.43 ± 0.15 |
| 3 | Unseen Real & Seen Fake Speakers | 5,557 | 100 (0, 100) | 13,871 | 10 (10, 0) | 0.01 ± 0.01 |
| 4 | Seen Real & Unseen Fake Speakers | 6,599 | 100 (100, 0) | 12,377 | 10 (0, 10) | 0.64 ± 0.06 |
| 5 | Mixed Seen / Unseen Speakers | 6,071 | 100 (50, 50) | 13,677 | 10 (5, 5) | 0.49 ± 0.05 |

Overall, the EERs across all five test settings are minimal, indicating that the model can detect deepfake-specific features rather than relying solely on speaker identity. Comparing Settings 1 and 2, where the distinction is whether speakers are seen or unseen during training, we observe only a slight increase in EER when speakers are unseen (0.06% to 0.43%). In Setting 3, where real speakers are unseen and fake speakers are seen, the model achieves almost perfect detection (0.01%), likely due to more fake data per speaker, though some speaker memorization may be occurring. In contrast, Setting 4, with seen real speakers and unseen fake speakers, results in a higher EER (0.64%), suggesting that the model struggles more with unseen fake speakers, possibly relying on learned fake speaker characteristics. Setting 5, with a mix of seen and unseen speakers, yields an EER of 0.49%, indicating better generalization than Setting 4, but still some performance drop with unseen fake speakers.

To conclude, the experiment demonstrates that the model can learn deepfake-specific features, but speaker identity does impact detection accuracy, particularly when encountering entirely unseen fake speakers.

## 5 LIMITATIONS AND FUTURE WORK

While SpeechFake provides a large and diverse dataset for speech deepfake detection, several limitations persist. Although the dataset includes 40 different speech generation tools, it does not cover all current or emerging techniques, as the field of speech generation evolves rapidly. Moreover, the variety of generation methods in the multilingual dataset is limited due to the scarcity of multilingual speech generation systems. Additionally, while SpeechFake offers a broad range of speaker voices and audio samples, the diversity in speaker identities and recording environments may still fall short of fully capturing real-world scenarios. For future work, we hope to potentially expand the dataset with more generation methods as they emerge, further enhancing its relevance and applicability for deepfake detection research.

## 6 CONCLUSION

In conclusion, SpeechFake addresses critical gaps in existing datasets for speech deepfake detection by providing a large-scale, diverse collection of over 3 million deepfakes generated using 40 different speech generation tools, spanning 46 languages. Through extensive experiments, we established baseline results and explored key factors such as generation methods, language diversity, and speaker variation, which significantly impact detection performance. Our findings emphasize the challenges of generalizing across unseen deepfakes while demonstrating the potential of Speech-Fake to drive future advancements in model robustness and generalization. We believe SpeechFake will be a valuable resource for the development of more sophisticated detection systems, helping to mitigate the risks associated with the growing threat of deepfake misuse.

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

# A  APPENDIX

In this appendix, we present additional information and analyses that complement the main paper. Section A.1 provides further details about the SpeechFake dataset. Section A.2 offers visualizations of deepfake audio samples, highlighting variations across generation methods, languages, and speakers. Finally, Section A.3 outlines the experimental settings and presents additional experimental results and analyses.

Table 7: The list of generation methods.

| No. | Method | Generator | Link |
|-----|--------|-----------|------|
| 1 | MelGAN (Kumar et al., 2019) | NV | https://github.com/kan-bayashi/ParallelWaveGAN |
| 2 | WaveGlow (Prenger et al., 2019) | NV | https://github.com/NVIDIA/waveglow |
| 3 | Parallel WaveGAN (Yamamoto et al., 2020) | NV | https://github.com/kan-bayashi/ParallelWaveGAN |
| 4 | HiFi-GAN (Kong et al., 2020) | NV | https://github.com/kan-bayashi/ParallelWaveGAN |
| 5 | Fullband-MelGAN (Yang et al., 2021) | NV | https://github.com/kan-bayashi/ParallelWaveGAN |
| 6 | StyleMelGAN (Mustafa et al., 2021) | NV | https://github.com/kan-bayashi/ParallelWaveGAN |
| 7 | FastDiff (Huang et al., 2022a) | NV | https://github.com/Rongjiehuang/FastDiff |
| 8 | BigVGAN (gil Lee et al., 2023) | NV | https://github.com/NVIDIA/BigVGAN |
| 9 | WaveNet (Van Den Oord et al., 2016) | TTS | https://github.com/r9y9/wavenet_vocoder |
| 10 | Tactotron2 (Shen et al., 2018) | TTS | https://github.com/NVIDIA/tacotron2 |
| 11 | Glow-TTS (Kim et al., 2020) | TTS | https://github.com/jaywalnut310/glow-tts |
| 12 | Grad-TTS (Popov et al., 2021) | TTS | https://github.com/huawei-noah/Speech-Backbones |
| 13 | FastSpeech2 (Ren et al., 2021a) | TTS | https://github.com/ming024/FastSpeech2 |
| 14 | PortaSpeech (Ren et al., 2021b) | TTS | https://github.com/keonlee9420/PortaSpeech |
| 15 | VITS (Kim et al., 2021) | TTS | https://github.com/jaywalnut310/vits/tree/main |
| 16 | StarGAN-VC (Li et al., 2021) | VC | https://github.com/yl4579/StarGANv2-VC |
| 17 | DiffGAN-TTS (Liu et al., 2022) | TTS | https://github.com/keonlee9420/DiffGAN-TTS |
| 18 | ProDiff-TTS (Huang et al., 2022b) | TTS | https://github.com/Rongjiehuang/ProDiff |
| 19 | EdgeTTS | TTS | https://github.com/rany2/edge-tts.git |
| 20 | TorToiSe (Betker, 2023) | TTS | https://github.com/neonbjb/tortoise-tts |
| 21 | StyleTTS2 (Li et al., 2024b) | TTS | https://github.com/yl4579/StyleTTS2 |
| 22 | OpenVoice (Qin et al., 2023) | VC | https://github.com/myshell-ai/OpenVoice |
| 23 | GPTSoVITS | VC | https://github.com/RVC-Boss/GPT-SoVITS |
| 24 | Fish Speech | TTS/VC | https://github.com/fishaudio/fish-speech |
| 25 | MeloTTS | TTS | https://github.com/myshell-ai/MeloTTS |
| 26 | ChatTTS | TTS | https://github.com/2noise/ChatTTS |
| 27 | CosyVoice (Du et al., 2024b) | TTS/VC | https://github.com/FunAudioLLM/CosyVoice |
| 28 | Parler-TTS (Lyth & King, 2024) | TTS | https://github.com/huggingface/parler-tts |
| 29 | FireRedTTS (Guo et al., 2024) | TTS | https://github.com/FireRedTeam/FireRedTTS |
| 30 | Seed-VC | VC | https://github.com/Plachtaa/seed-vc |
| 31 | Volcengine API | TTS | https://www.volcengine.com |
| 32 | Baidu API | TTS | https://cloud.baidu.com |
| 33 | AliYun API | TTS | https://www.aliyun.com |
| 34 | Xfyun API | TTS | https://www.xfyun.cn |
| 35 | Moyin API | TTS | https://www.moyin.com |
| 36 | Microsoft API | TTS | https://azure.microsoft.com |
| 37 | Google API | TTS | https://cloud.google.com |
| 38 | Amazon API | TTS | https://docs.aws.amazon.com/polly |
| 39 | OpenAI API | TTS | https://platform.openai.com |
| 40 | GPT4o API | TTS | https://platform.openai.com |

## A.1  DATASET DETAILS

**Generation Methods**  Table 7 provides a comprehensive list of the 40 speech generation tools used to create the SpeechFake dataset. These tools cover a broad range of techniques, including TTS, VC, and NV systems. Some methods, such as Fish Speech and CosyVoice, can be used for multiple generation tasks (e.g., TTS and VC), demonstrating the versatility of these systems.

**License**  For the 30 open-source tools, we carefully reviewed their licenses to ensure compliance with the construction and release of a publicly available dataset. Most of the data generated using these tools is released under CC BY-NC 4.0, with certain portions, such as data from SeedVC, licensed under GPL-3.0 to comply with its requirements. The remaining 10 generation tools are commercial APIs, for which we obtained paid access, ensuring compliance with non-commercial research usage policies.

**Metadata**  SpeechFake provides detailed metadata for each generated speech sample, including:

- **Basic Labels**: Identifying real or fake speech.
- **Generation Method**: Specifying the tool used to create the speech.

- **Speaker Information**: Providing identity labels for the real speaker or the generated voice.
- **Language ID**: Indicating the language of the audio sample.
- **Text Transcriptions**: Providing the corresponding text for the generated speech.

**Additional Use Cases**  While SpeechFake is primarily designed for traditional speech deepfake detection, the detailed metadata it provides opens up various opportunities for other research areas. Below, we outline some additional use cases for the dataset:

- **Deepfake Detection with Privacy Preservation**: SpeechFake can support research into detecting deepfakes while maintaining privacy protections for both speaker identities and sensitive content. Techniques such as federated learning, differential privacy, or privacy-preserving machine learning can be explored to allow models to detect deepfakes without exposing or compromising personal data. This opens up possibilities for safer deployment of detection models in real-world applications where privacy is a concern.
- **Automatic Speech Recognition (ASR)**: The synthetic speech in SpeechFake provides a valuable opportunity to assess how ASR systems handle artificially generated audio. Researchers can explore whether incorporating deepfake speech can enhance ASR performance by training models to be more robust against synthetic variations, thereby improving accuracy and adaptability in handling diverse, challenging speech inputs.
- **Security and Adversarial Audio Attacks**: Synthetic speech from SpeechFake can be utilized in adversarial attacks aimed at ASR systems or voice-activated applications. The dataset offers a resource for developing and testing countermeasures against adversarial audio manipulations, enabling researchers to fortify voice-driven systems against security threats, ensuring their reliability and resilience in real-world scenarios.

Table 8: Metadata comparison across various speech deepfake datasets. The "-" symbol in Language column indicates datasets that do not support multi-language data.

| Dataset | Label | Generator | Speaker | Language | Text |
|---|---|---|---|---|---|
| ASVspoof2015 (Wu et al., 2014) | ✓ | ✓ | ✓ | - | ✗ |
| FakeOrReal (Reimao & Tzerpos, 2019) | ✓ | ✗ | ✗ | - | ✗ |
| ASVspoof2019-LA (Nautsch et al., 2021) | ✓ | ✓ | ✓ | - | ✗ |
| WaveFake (Frank & Schönherr, 2021) | ✓ | ✓ | ✓ | ✓ | ✗ |
| ASVspoof2021-LA (Yamagishi et al., 2021) | ✓ | ✗ | ✗ | - | ✗ |
| ASVspoof2021-DF (Yamagishi et al., 2021) | ✓ | ✗ | ✗ | - | ✗ |
| ADD2022 (Yi et al., 2022) | ✓ | ✗ | ✗ | - | ✗ |
| CFAD (Ma et al., 2024) | ✓ | ✓ | ✗ | - | ✗ |
| In-the-Wild (Müller et al., 2022) | ✓ | - | ✓ | - | ✗ |
| ADD2023 (Yi et al., 2024) | ✓ | ✗ | ✗ | - | ✗ |
| HABLA (Tamayo Flórez et al., 2023) | ✓ | ✓ | ✓ | - | ✗ |
| MLAAD (Müller et al., 2024) | ✓ | ✓ | ✓ | ✓ | ✓ |
| CD-ADD (Li et al., 2024c) | ✓ | ✓ | ✗ | - | ✗ |
| VoiceWukong (Yan et al., 2024) | ✓ | ✓ | ✗ | ✓ | ✗ |
| DFADD (Du et al., 2024a) | ✓ | ✓ | ✗ | - | ✗ |
| CVoiceFake (Li et al., 2024a) | ✓ | ✓ | ✗ | ✓ | ✓ |
| SpeechFake | ✓ | ✓ | ✓ | ✓ | ✓ |

## A.2 AUDIO SAMPLES

In this section, we provide visualizations of the waveform and mel-spectrogram for the generated speech, highlighting the significant variations across different generation methods, languages, and speakers.

**Generator**  Figure 5 compares real audio with eight speech deepfakes generated by cutting-edge speech generation methods. Despite the same speech content, the generated audio exhibits differences in speed, rhythm, accent, and background noise. For example, variations in speed and rhythm can be observed through the density and spacing of peaks in the waveform, while background noise introduces irregular, low-amplitude fluctuations. Additionally, the mel-spectrograms for each audio sample display variations in energy distribution (intensity across frequency bands, visible as brightness), frequency patterns (harmonic structures and their arrangement), and clarity (sharpness of

features, with blurriness indicating potential distortions), indicating how different generation techniques shape the acoustic properties of the speech. These differences underscore the challenges in detecting deepfakes, as the synthetic speech can vary significantly depending on the generation method used.

**Language** Figure 6 presents generated audio samples for nine different languages, all created using the same generation method (EdgeTTS). While the language content varies—affecting factors such as speech speed, rhythm, and intonation—the mel-spectrograms display consistent global patterns. For example, the overall energy distribution across frequency bands and the structural arrangement of harmonics remain similar, reflecting the uniform processing applied by the EdgeTTS model. However, linguistic differences, such as unique intonation patterns and phoneme timing, introduce subtle variations in the finer details of the spectrograms. These observations highlight the influence of language-specific traits on acoustic features, while also demonstrating the model's consistent generation approach across languages.

**Speaker** Figure 7 illustrates the generated audio for nine different voices, produced by a single generation method (TorToiSe) using the same text input. The choice of speaker introduces variations in the acoustic properties of the speech, reflected in features such as pitch, timbre, and pacing. In the mel-spectrograms, pitch is represented by the spacing between harmonic bands, timbre by the energy distribution across frequency bands, and pacing by the density of temporal transitions. These variations highlight the model's ability to modify voice-specific traits while keeping the content identical, showing how speaker identity influences the acoustic properties of the generated speech.

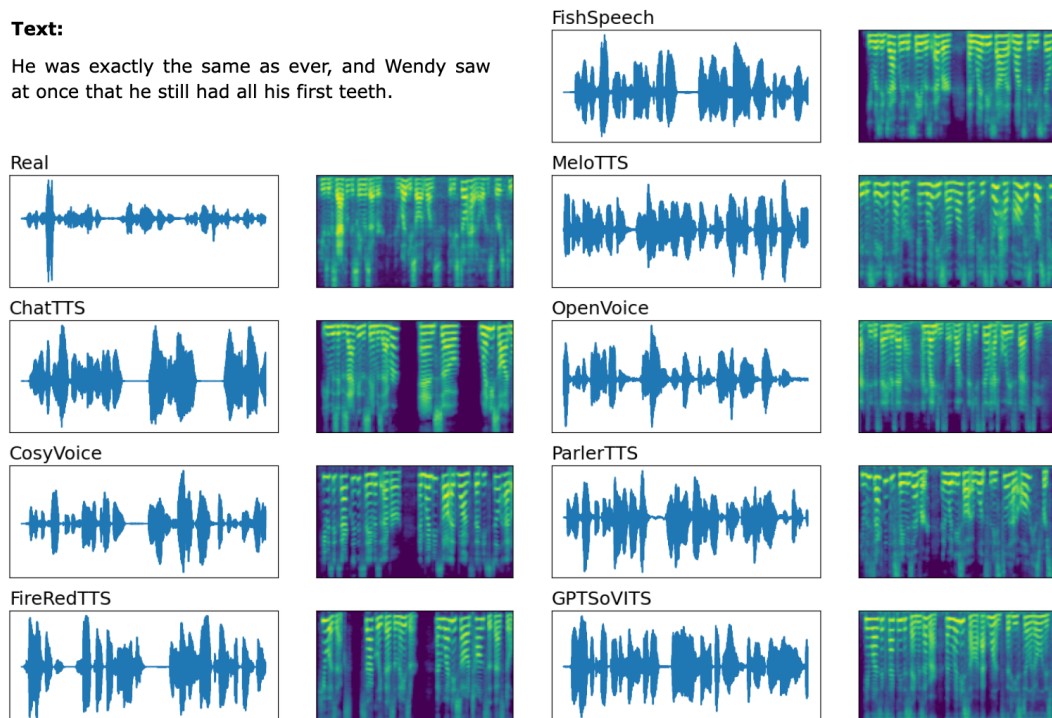

Figure 5: Visualization of waveform and mel-spectrograms for a single text input across different generation methods.

## A.3 EXPERIMENT DETAILS

**Experimental Settings** Table 9 outlines the training configurations for the two state-of-the-art models used in our experiments. The basic settings are consistent with the training setup proposed by Tak et al. (2022). Unlike previous research on deepfake detection, we opted not to apply data

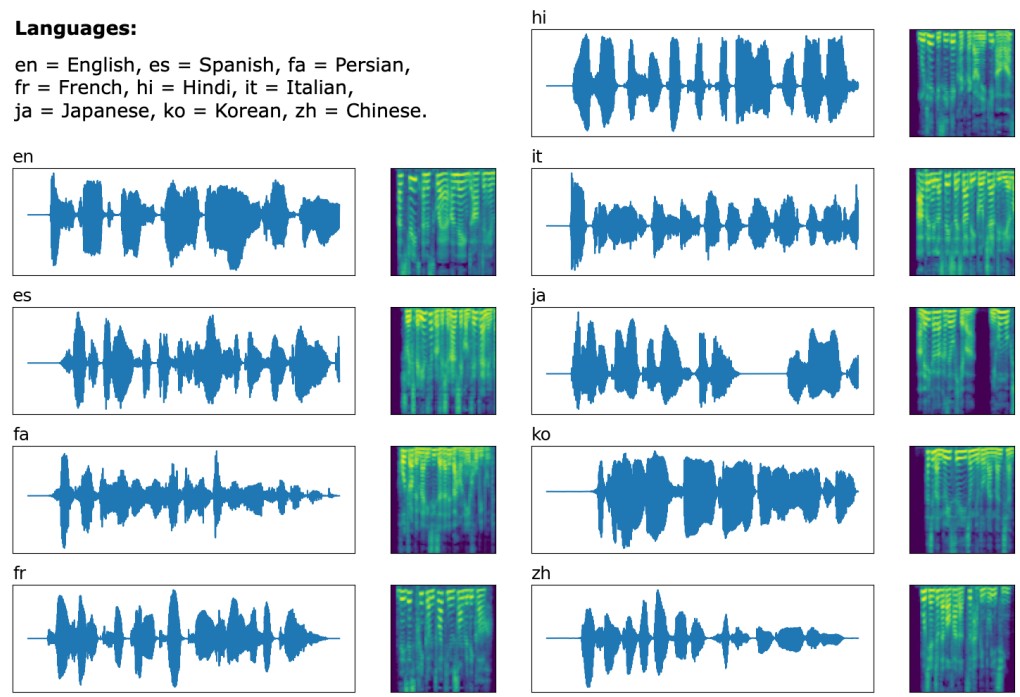

Figure 6: Visualization of waveform and mel-spectrograms for speech deepfakes in different languages, generated by EdgeTTS.

augmentation in order to isolate the fundamental effects of the audio data and avoid potential biases introduced by augmentation methods, which may not generalize well across all datasets. Given the imbalance between deepfake and real samples, we employed weighted cross-entropy loss to ensure balanced training. Both models were trained for 50 epochs on 8 A100 GPUs.

Table 9: Training Details of AASIST and W2V+AASIST Models

| Training Parameter | AASIST | W2V+AASIST |
|---|---|---|
| Input Audio | Chunk or pad to 4s | Chunk or pad to 4s |
| Data augmentation | None | None |
| Optimizer | Adam | Adam |
| Learning Rate | 1e-4 | 1e-6 |
| Weight Decay | 1e-4 | 1e-4 |
| Batch Size | 1024 | 512 |
| Total Epochs | 50 | 50 |
| Loss Function | Weighted Cross Entropy (0.9 for real, 0.1 for fake) | |

Table 10: Performance evaluation (EER%) of different models trained on ASVspoof2019 (ASV19) or Bilingual Dataset (BD) across multiple test sets.

| Training Dataset | Model | Testing Dataset | | | | | | | | | |
|---|---|---|---|---|---|---|---|---|---|---|---|
| | | ASV19 | FOR | WF | ASV21LA | ASV21DF | CFAD | ITW | MLAAD | CD-ADD | ASV24 |
| ASV19 | AASIST | 1.88 | 36.08 | 21.17 | 7.30 | 19.32 | 43.95 | 45.27 | 15.34 | 49.53 | 41.89 |
| BD | | 23.62 | 23.35 | 4.30 | 32.56 | 21.68 | 34.32 | 7.53 | 22.97 | 22.52 | 35.02 |
| BD-EN | | 30.65 | 28.99 | 8.54 | 49.75 | 34.29 | 43.39 | 6.96 | 27.16 | 23.24 | 40.82 |
| BD-CN | | 16.56 | 25.48 | 5.88 | 21.19 | 23.20 | 32.34 | 8.54 | 30.15 | 39.75 | 34.39 |
| ASV19 | W2V+AASIST | 0.89 | 6.18 | 3.48 | 6.57 | 2.98 | 20.53 | 10.07 | 18.26 | 8.55 | 1.41 |
| BD | | 2.91 | 6.00 | 0.58 | 7.27 | 2.85 | 12.39 | 2.01 | 12.86 | 2.42 | 0.71 |
| BD-EN | | 5.28 | 8.33 | 0.96 | 11.91 | 2.97 | 21.42 | 2.62 | 16.75 | 3.54 | 0.71 |
| BD-CN | | 0.99 | 4.88 | 0.64 | 3.92 | 5.87 | 11.72 | 3.34 | 10.17 | 7.16 | 1.17 |

**Performance on Other Datasets** We extend the evaluation from Table 3 to include 10 additional datasets beyond our primary test set. These datasets are ASVspoof2019 (ASV19), FakeOr-

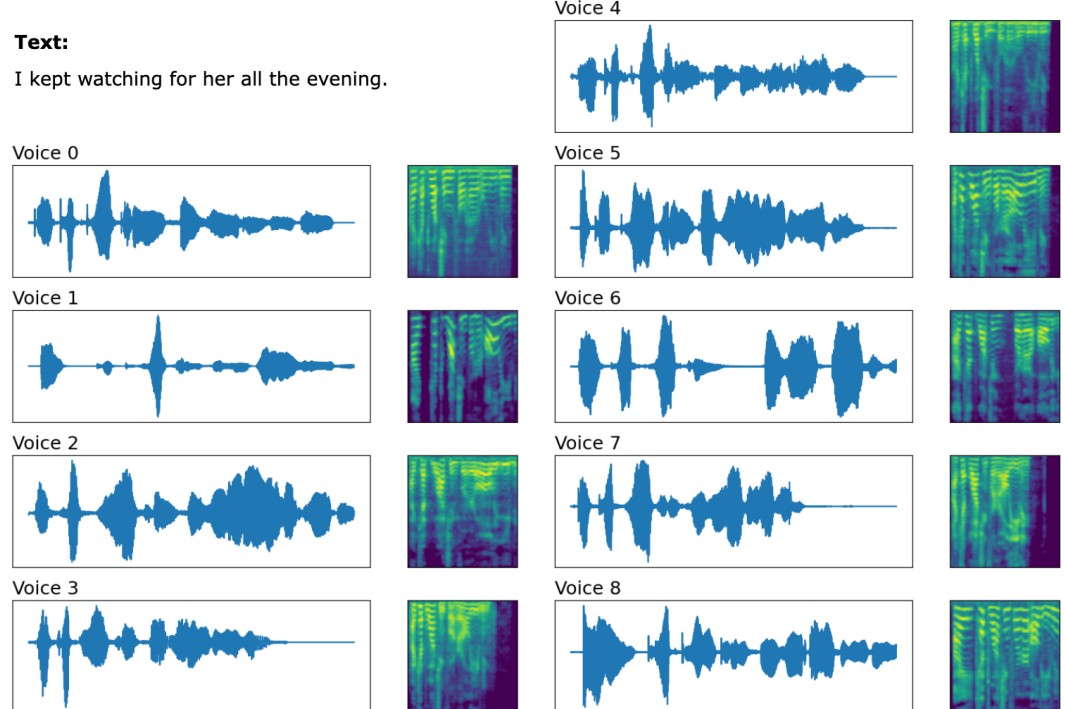

Figure 7: Visualization of waveform and mel-spectrograms for a single text input across different generated voices, generated by TorToiSe.

Real (FOR), WaveFake (WF), ASVspoof2021-LA (ASV21LA), ASVspoof2021-DF (ASV21DF), CFAD, In-The-Wild (ITW), MLAAD, CD-ADD, and ASVspoof5 (ASV24). For ASV21LA, ASV21DF, CFAD, and CD-ADD, evaluations are conducted on their original test sets. For ASV24, the development set is used as labels for the evaluation set are unavailable. For MLAAD and WF, which don't have predefined train/test splits, we randomly select 15,000 clips for evaluation.

For most of these test sets, models trained on SpeechFake consistently outperform those trained on ASV19. This demonstrates the effectiveness of SpeechFake in capturing diverse and challenging spoofing scenarios. However, some exceptions are observed with the ASVspoof datasets from 2019 and 2021, as these datasets share a more similar distribution with the ASV19 training data, giving models trained on ASV19 an advantage in these specific cases.

An additional observation is that models trained on the full BD dataset do not always achieve the best performance compared to those trained on its subsets, BD-EN and BD-CN. This may be attributed to the distinct generation methods used in BD-EN and BD-CN, which introduce either similar attributes or significant differences relative to the test sets. Such variations in distribution can result in models trained on BD-EN or BD-CN achieving better alignment with certain test sets, while the broader BD dataset, encompassing more diverse data, may dilute this alignment in some cases.

**Score Distribution** We provide visualizations of the score distributions for various training and testing configurations from our experiments in Section 4.2 and Section 4.3.

Figure 8 shows the score distribution for models trained on the full BD dataset as well as its English (BD-EN) and Chinese (BD-CN) subsets. For the model trained on the full BD set, the scores for real and fake samples are clearly separable, corresponding to the low EER across these test sets. However, when models are trained on either BD-EN or BD-CN, we observe some overlap in the score distributions when tested on the complementary language set. This fusion occurs primarily due to the model's difficulty in generalizing across different languages, particularly when distinguishing fake samples in an unfamiliar linguistic context.

Figure 9 presents the score distribution for models trained on the BD TTS, NV, and VC subsets. Each model performs best on its corresponding test set (diagonal), but shows degraded performance on the other two sets containing unseen deepfakes. Unlike in the previous figure, the overlap in score distributions here primarily reflects the difficulty in distinguishing real samples from fake ones generated by unseen techniques. This suggests that models trained on specific generation methods (TTS, NV, or VC) may struggle to generalize to different types of deepfake generation techniques, highlighting the challenge of cross-generator generalization.

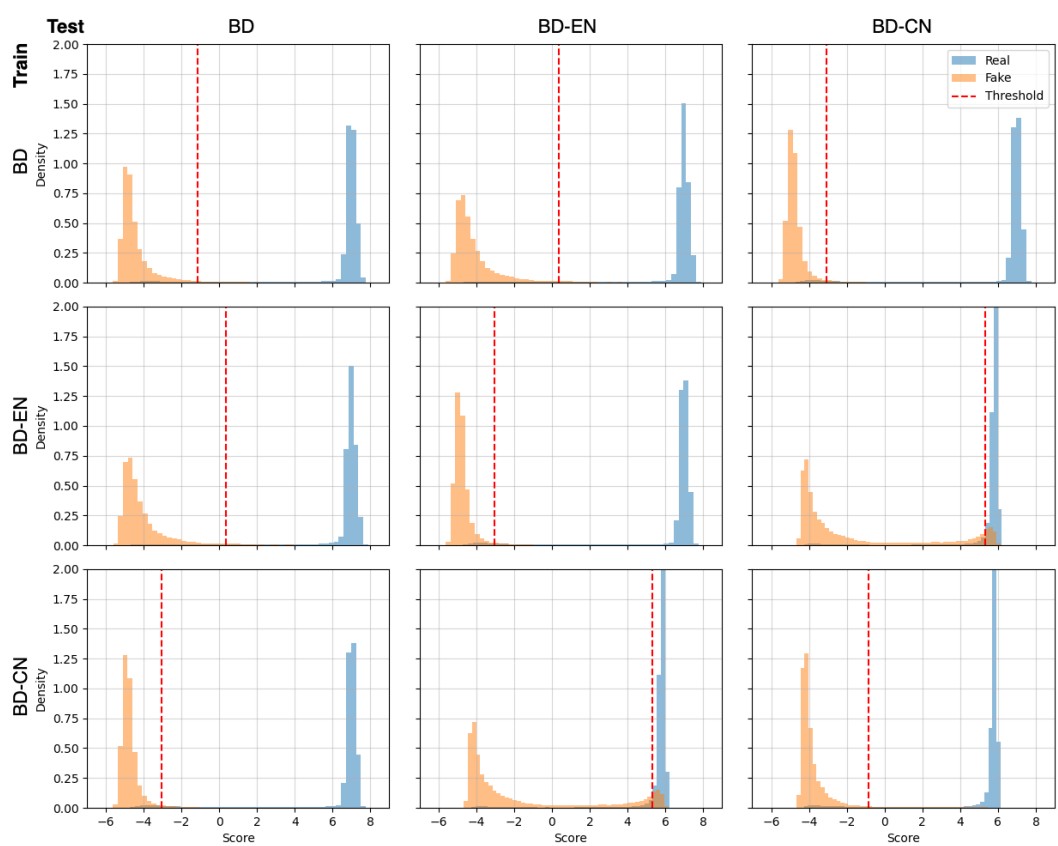

Figure 8: Score distribution for real and fake samples when training on BD, BD-EN, and BD-CN, tested on their respective test sets using the W2V+AASIST model. The EER values correspond to those in Table 3, with the indicated threshold representing the EER threshold.

**Cross-model Evaluation**   To complement the cross-generator performance evaluation in Section 4.3, where tests were conducted across different generator types, and to assess the impact of data from the latest generation methods, we perform additional evaluations using models trained on individual methods. As shown in Figure 10, the EER remains minimal when testing on the corresponding test sets for each method, but there is a significant degradation in performance when tested on other methods' test sets. In certain cases, some models demonstrate relatively good results on specific test sets (e.g., the FireRedTTS-trained model on the ChatTTS test set), but these results are inconsistent across all test sets. This further emphasizes the challenge of generalization to unseen deepfakes, highlighting the need for more robust detection models capable of adapting to a wider range of generation methods.

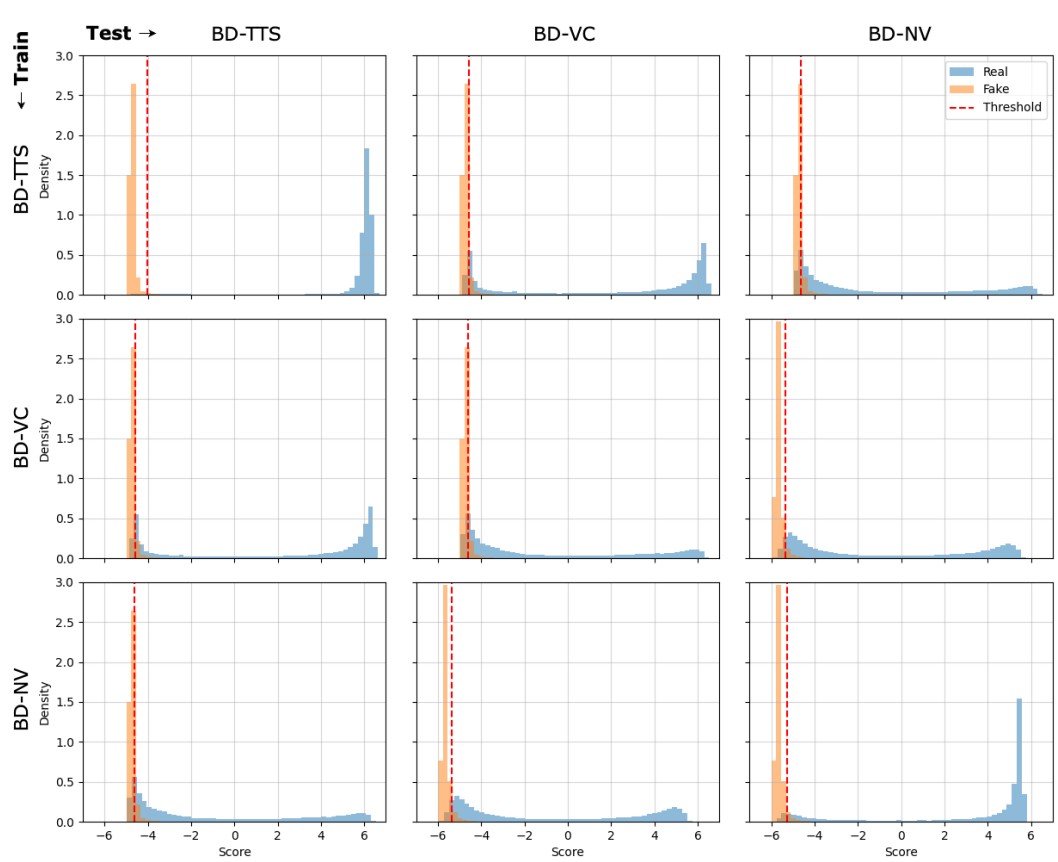

Figure 9: Score distribution for real and fake samples when training on BD-TTS, BD-VC, and BD-NV, tested on their respective test sets using the W2V+AASIST model. The EER values correspond to those in Table 4, with the indicated threshold representing the EER threshold.

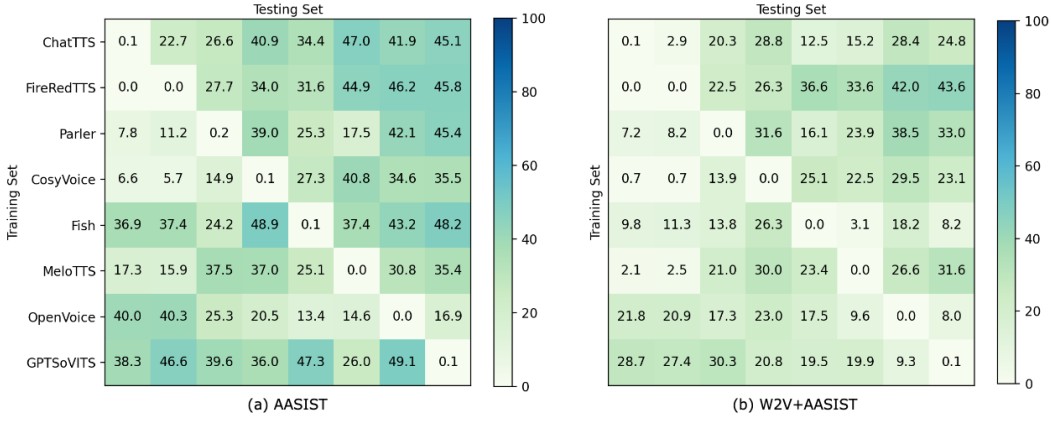

Figure 10: Cross-evaluation performance (EER%) of different generation models as training sets across their testing sets.

