# OpenReview forum: "SpeechFake: A Large-Scale Multilingual Speech Deepfake Dataset Toward Cutting-Edge Speech Generation Methods"
_ICLR.cc/2025/Conference — Submitted to ICLR 2025_

### Official Review · Reviewer_EVny · 2024-11-03

**Soundness:** 2
**Presentation:** 3
**Contribution:** 2
**Rating:** 5
**Confidence:** 4

**Summary:**

The paper introduces a new dataset, SpeechFake, for training and evaluating speech deepfake detectors.  The dataset is larger than previous such datasets, and includes deepfakes generated by a larger number (40) of speech generation systems, including text-to-speech (TTS), voice conversion (VC), and neural vocoders (NV).  The paper also provides baseline results using existing deepfake detection models, demonstrating that training the detectors on SpeechFake is more effective across multiple evaluation sets than training on a previous such dataset, ASV19, and measuring the detection performance across speakers, languages, and deepfake generation methods.

**Strengths:**

+ The paper addresses an important problem.

+ The study of performance across multiple dimensions (languages, speakers, seen/unseen generation methods) is valuable.

+ The dataset will probably be useful for research beyond deepfake detection, like data augmentaiton.

**Weaknesses:**

- The paper doesn't include enough empirical comparisons to make a convincing case for the value added of the new dataset.  It includes a comparison of training on the new dataset vs. on ASV19, but not vs. other existing datasets, or vs. combinations of existing datasets, which might provide a similar generalization benefit.

- The contribution is a bit narrow for ICLR.  A dataset + baseline paper like this would be a better fit in a speech conference or related workshop.

- The description of dataset construction needs more detail, e.g. how multiple speaker IDs were sampled and generated with each model.  Ideally the paper would include an "algorithm" for sampling examples using each method.

**Questions:**

- How is the quality filtering done?  The text states "generated speech with noticeable distortions, excessive noise, or unnatural artifacts is discarded".  Is this done automatically, and if so how?

- The detection performance results in Tables 5 and 6 are much better for all settings than in Tables 3 and 4.  Why is that?  For example, why are the English and Chinese results in Table 4 much better than those in the BD-EN and BD-CN columns of Table 3?

- The extremely low EERs in Table 5 suggest that the multilingual dataset is too easy and has very limited value for deepfake detection research.  Is this a misunderstanding of the results?  Please correct me if I'm missing something.

- For the settings with very low EERs, it would be helpful to include confidence intervals or significance estimates.  It is hard to tell whether we can really draw conclusions based, for example, on the differences between speaker settings in Table 6.

More minor details:

- The title is a bit hard to parse.  I think the word "toward" should probably be something else, e.g. "generated using".

- The third contribution in the bullet list (at the end of the introduction) doesn't seem like an independent contribution (more like an elaboration of the others).

- In Table 1, what does "-" mean?

- The first sentence in Related Work suggests that TTS and VC are the only two speech generation tasks (though NV is added on later).  However, there are other tasks like speech content editing, style conversion, and enhancement.

- The related work sub-section on existing datasets doesn't explicitly state what is new/different about the proposed one (though this is briefly stated elsewhere).

- In the appendix, what does "speaker information" include?  The description is "details about the speaker or the generated voice identity", but I'm not sure what this means.

- In the appendix, the figures and surrounding descriptions comparing waveforms/mel spectrograms are unclear and make questionable statements (as far as I can tell) about the examples shown.  For example, A.2 states that the mel spectrograms show "variations in energy distribution, frequency patterns, and clarity".  What exactly is meant by this?  Another similar statement is "the resulting mel-spectrograms share some common characteristics, particularly in their overall structure and energy distribution across frequencies".  What does "overall structure" and "energy distribution" mean, and how do you measure that they are similar here vs. in other examples across multiple speakers?  Yet another statement of this type is about "... distinct characteristics, such as pitch, timbre, and pacing, which are clearly reflected in the waveforms and mel-spectrograms".  How do you measure the timbre in the waveforms or mel spectrograms?  On a more minor note, labeling the axes in the waveform and spectrogram plots would help.

**Details Of Ethics Concerns:**

There may be no violation here, but I am a bit concerned that the paper doesn't fully specify the license that will be used when releasing the dataset publicly.  The appendix only states that the authors have checked the licenses of the deepfake generation software that they used, and thus ensured that the dataset "complies with non-commercial research usage policies".  Considering that the dataset contains deepfakes of real people (if I've understood correctly), I think more detail is needed about how the authors ensure this and how the data will be licensed.

---

> ### Author Response · Authors · 2024-11-20
> **Response to Official Review of Submission14000 by Reviewer EVny - Part 1**
>
> We sincerely thank the reviewer for their thorough evaluation of our paper and for providing insightful comments and constructive feedback. Below, we address each of the points raised.
>
> **W1: Empirical Comparisons with Other Datasets**
>
> In our initial experiment, we report baseline results for models trained on ASV19, as it is a widely used setup for this task, and many existing datasets are too small to serve as training sets and are primarily used for testing. Results for models trained on other datasets will be included in the next revision of the paper, as thorough training and evaluation require significant computational resources and time. Meanwhile, we have added results for models trained on ASV19 and SpeechFake and tested on additional test sets in Table 10 of the revised paper, with a partial table provided here. These evaluations span 10 test sets, covering both earlier and more recent datasets. For most test sets, models trained on SpeechFake consistently outperform those trained on ASV19, demonstrating its effectiveness in capturing diverse and challenging spoofing scenarios.
>
> Table 10*: Performance evaluation (EER\%) of model (W2V+AASIST) trained on ASVspoof2019 (ASV19) or Bilingual Dataset (BD) across multiple test sets.
> | Training Dataset | ASV19  | FOR   | WF    | ASV21LA | ASV21DF | CFAD  | ITW   | MLAAD | CD-ADD | ASV24  |
> |------------------|--------|-------|-------|---------|---------|-------|-------|-------|--------|--------|
> | ASV19           | 0.89   | 6.18  | 3.48  | 6.57    | 2.98    | 20.53 | 10.07 | 18.26 | 8.55   | 1.41   |
> | BD              | 2.91   | 6.00  | 0.58  | 7.27    | 2.85    | 12.39 | 2.01  | 12.86 | 2.42   | 0.71   |
> | BD-EN           | 5.28   | 8.33  | 0.96  | 11.91   | 2.97    | 21.42 | 2.62  | 16.75 | 3.54   | 0.71   |
> | BD-CN           | 0.99   | 4.88  | 0.64  | 3.92    | 5.87    | 11.72 | 3.34  | 10.17 | 7.16   | 1.17   |
>
> **W2: Narrow Contribution**
>
> We appreciate the reviewer’s observation and would like to clarify that our work extends beyond simply presenting a dataset and baseline results. In addition to establishing baselines, we conduct detailed analyses that explore critical factors influencing deepfake detection performance, such as generation methods, language diversity, and speaker variation. These analyses provide new insights into the challenges of generalization, a key issue in deepfake detection that transcends specific domains like speech.
>
> **W3: Dataset Construction Details**
>
> We have added more details about the generation process in Section 3.1 of the revised paper. Regarding the selection of multiple speaker IDs: For TTS systems, we use the voices provided by each method and balance the selection to include a mix of genders whenever possible. For VC systems, we sample reference voices from real datasets, ensuring diversity and representativeness. For NV systems, the generated speech retains the same voice characteristics as the original input audio from the real datasets.
>
> **Q1: Quality Filtering**
>
> The detection of low-quality sounds is performed through human verification. Before the batch generation, we generate over 100 diverse sample audio clips for each system. Systems producing artifacts, silences, or other issues were filtered out at this stage. Only after passing this filtering process did we proceed with batch generation to create data covering different languages and voices. After generation, we applied voice activity detection (VAD) to filter out speech segments with less than 0.5 seconds of active speech. Additionally, random samples from each system were manually reviewed throughout the process to ensure consistency and naturalness.
>
> **Q2: Different Detection Performance**
>
> The differences in detection performance across Tables 3, 4, 5, and 6 are due to variations in training and testing settings, as described in each experiment section:
>
> - Tables 3 and 4 present results on the BD (Bilingual Dataset), using different subsets for training or testing to evaluate overall performance (Table 3) or cross-generator performance (Table 4).
> - Table 5 focuses on the MD (Multilingual Dataset), which uses entirely separate train, dev, and test sets from MD instead of BD, with differences in languages and generation methods.
> - Table 6 evaluates a smaller dataset selected from BD, using a single generation method but different generated voices to study cross-speaker performance.
>
> The improved results in Tables 5 and 6 are primarily due to the smaller test sets and fewer generation methods covered, which make the evaluation less challenging compared to the broader and more diverse settings in Tables 3 and 4.

---

> ### Author Response · Authors · 2024-11-20
> **Response to Official Review of Submission14000 by Reviewer EVny - Part 2**
>
> **Q3: Performance on Multilingual Dataset**
>
> The low EERs in Table 5 result from smaller test sets and fewer generation methods, making the tests less challenging than the broader BD sets. AASIST, without pretraining, shows suboptimal performance on unseen languages (e.g., EER = 9.57% for Hindi, 4.90% for French), highlighting the dataset’s challenges. The lower EERs for W2V+AASIST reflect the advantage of Wav2Vec XLSR’s multilingual pretraining, demonstrating the dataset’s utility in evaluating models across language mismatches and unseen conditions.
>
> However, it is inaccurate to conclude that the multilingual dataset has limited value. Without a multilingual pretrained model, the language gap presents significant challenges for detection. Even with such a pretrained model, testing on the complete dataset (including the hidden portion) for each language can result in reduced performance, as demonstrated in the table below.
>
> Table R3: EER (%) results for each language across the entire dataset, using W2V+AASIST models trained on English and Chinese.
> | en   | zh   | es   | fr    | hi    | ja   | ko   | fa   | it   | others |
> |------|------|------|-------|-------|------|------|------|------|--------|
> | 0.27 | 1.39 | 7.98 | 12.08 | 11.90 | 5.14 | 2.54 | 4.42 | 3.48 | 4.64   |
>
> **Q4: Confidence Intervals for Low EERs**
>
> We appreciate the reviewer’s suggestion to include confidence intervals for settings with very low EERs. In the updated Table 6, we have added confidence intervals to all EER values to provide a clearer understanding of the results. The updated table also shows that the conclusions remain consistent with the original analysis.
>
> Table 6*: EER(%) results of cross-speaker testing trials.
> | No. | Test Setting                     | EER(%)                  |
> |-----|----------------------------------|-------------------------|
> | 1   | Seen Real & Fake Speakers        | 0.06 ± 0.01            |
> | 2   | Unseen Real & Fake Speakers      | 0.43 ± 0.15            |
> | 3   | Unseen Real & Seen Fake Speakers | 0.01 ± 0.01            |
> | 4   | Seen Real & Unseen Fake Speakers | 0.64 ± 0.06            |
> | 5   | Mixed Seen / Unseen Speakers     | 0.49 ± 0.05            |
>
>
> **Q5: About the Title**
>
> We chose the word “toward” to emphasize the dataset’s role in advancing speech deepfake detection by incorporating more advanced speech generation methods. While our dataset includes cutting-edge techniques, it also retains classical methods to ensure comprehensive coverage. Using “generated using” would inaccurately suggest the dataset exclusively features cutting-edge methods.
>
> **Q6: About the Contribution**
>
> We appreciate the reviewer’s observation regarding the third contribution. To address this, we have rephrased it to emphasize its distinct value: “SpeechFake provides a valuable resource for developing robust deepfake detection models, demonstrating superior performance on existing datasets. It also supports future research in improving model generalization and advancing detection strategies for emerging techniques.” The revised contribution now highlights SpeechFake’s immediate utility in demonstrating superior performance on existing datasets and its long-term support for advancing future research, such as improving model generalization and detection strategies. In contrast, the first contribution focuses on introducing the dataset’s scale, diversity, and features, while the second emphasizes the experiments and analyses conducted.
>
> **Q7: Symbol “-” in Table 1**
>
> The “-” in Table 1 indicates that the dataset does not specify the number of speakers or generators included. We have clarified this in the table caption for the revised version of the paper.
>
> **Q8 & 9: Description in related work**
>
> For the first part of the Related Work section, Speech Generation, the sentence “Speech generation, or speech synthesis, can be broadly divided into two main tasks: Text-to-Speech (TTS) and Voice Conversion (VC)” does not imply that these are the only speech generation tasks. Rather, it highlights TTS and VC as the primary tasks in this domain. Other tasks, such as speech content editing and style conversion, often build upon these foundational tasks and share similar methods with modifications tailored to their specific objectives. Regarding Neural Vocoder (NV), it is not a standalone task but a fundamental component of many speech generation methods. Since vocoded speech plays a significant role in detecting deepfakes, we included such data in our dataset to enhance its utility.
>
> For the second part, Speech Deepfake Datasets, the differences between our dataset and existing ones are stated in the Introduction. To make this clearer, we have added an additional description in the revised version of the paper to explicitly highlight these differences within the Related Work section.

---

> ### Author Response · Authors · 2024-11-20
> **Response to Official Review of Submission14000 by Reviewer EVny - Part 3**
>
> **Q10: About “Speaker Information”**
>
> We have revised the description to “Providing identity labels for the real speaker or the generated voice.” for clearer expression.
>
> **Q11: Unclear statement for description of waveform / mel**
>
> Thank you for your feedback regarding the figures and descriptions in the appendix. Below, we briefly clarify the terms for the unclear parts.
>
> For waveforms:
>
> - Speed and Rhythm: Reflected in the density and spacing of peaks, with faster speech showing more densely packed peaks and slower speech wider spacing.
> - Accent and Intonation: Variations in peak amplitude indicate stress or emphasis, characteristic of accents or intonation patterns.
> - Background Noise: Irregular, low-amplitude fluctuations suggest noise, while smoother patterns indicate cleaner audio
>
> For mel-spectrograms:
>
> - Energy Distribution: Represented by brightness across frequency bands.
> - Frequency Patterns: Harmonic structures show pitch (spacing between harmonics) and stability (consistency of patterns).
> - Clarity: Defined by sharpness of features; blurriness indicates noise or distortion.
>
> We have updated the descriptions in the revised paper for improved clarity.
>
> **Ethics Review: Privacy, security and safety**
>
> To clarify, the dataset does not contain deepfakes of identifiable real people. Regarding the voice, it may originate from training data (TTS), reference voice (VC), or original audio (NV).
>
> - From training data: the TTS systems used are typically trained on publicly available datasets that do not include identifiable speaker information. In cases where systems are trained on private data, we can only select voices from the output without any connection to real individuals.
> - From reference voice or original audio: The voices for VC and NV systems are similarly taken from publicly available speech datasets without any identifiable speaker information.
>
> For the content, all text or speech samples are taken from publicly available speech datasets commonly used in speech generation research. These datasets do not contain harmful or sensitive content.
>
> Besides, publicly released datasets in prior work follow a similar approach, using open-source tools and public speech datasets for generating deepfake data. These datasets have not been associated with privacy or security concerns.
>
> The dataset will primarily be released under CC BY-NC 4.0, with certain portions licensed under GPL-3.0 to comply with the requirements of specific tools.

---

> > ### Comment · Reviewer_EVny · 2024-11-30
> >
> > Thank you for the detailed responses.  The clarifications about details of the dataset are helpful, and adding them to the paper improves its quality and usefulness.  My overall rating remains unchanged, mainly because I believe the contribution is a bit too narrow for ICLR and would fit better in a speech venue.  To respond briefly to a couple of the points:
> >
> > Q3: Performance on Multilingual Dataset
> > The low EERs in Table 5 result from smaller test sets and fewer generation methods, making the tests less challenging than the broader BD sets.  ...
> >
> > --> Why would a smaller test set be less challenging?  It should result in higher-variance results, but not necessarily better results.  This seems like a basic statistical property, but please let me know if I may be missing something.
> >
> > Q11: Unclear statement for description of waveform / mel
> > Thank you for your feedback regarding the figures and descriptions in the appendix. Below, we briefly clarify ...
> >
> > --> Thanks for the clarifications of the terms.  I am familiar with how properties like speed, intonation, and so on affect waveforms and spectrograms, but I still do not believe that your examples demonstrate the implied differences.  Overall, I feel that these descriptions detract from rather than enhance the impact of the paper.

---

### Official Review · Reviewer_cq2e · 2024-11-04

**Soundness:** 3
**Presentation:** 3
**Contribution:** 3
**Rating:** 6
**Confidence:** 3

**Summary:**

The paper introduces SpeechFake, a large-scale multilingual dataset designed explicitly for speech deepfake detection. It offers a vast collection of over 3 million samples (amounting to 3,000 hours of audio) generated using 40 advanced speech synthesis tools across 46 languages. The paper provides a comprehensive baseline evaluation for SpeechFake, analyzing factors like generation methods, language diversity, and speaker variation.

**Strengths:**

- SpeechFake Dataset: A large-scale and diverse dataset for speech deepfake detection, encompassing a wide range of speech generation methods. The dataset consists of 3 million utterances, totalling 3,000 hours. It would be beneficial if the authors open-source the entire dataset (including hidden), rather than just a subset.

**Weaknesses:**

- Multilingual Focus: While the work prioritizes multilingualism, it primarily focuses on English and Chinese. Training datasets are lacking for 37 out of the 46 languages, with only a set of 5,000 utterances released for testing purposes for each of these 37 languages.
- Coverage of Speech Generation Tools: Although the dataset includes 40 different speech generation tools, it does not encompass all current techniques, nor does it provide a way to account for future techniques that may emerge. This limitation suggests that the dataset may not remain fully relevant over time and could require updates or recreation as the field advances.

**Questions:**

- Generation Process Details: It would be helpful if the authors could clarify how they determine the number of utterances generated using different speech generation tools for a particular original utterance.
- Choice of Test Sets (Table 3): Referring to Table 3, could the authors provide insights into why test sets from 2019 were selected rather than more recent versions? In fact, evaluating on all available benchmarks might have provided a more comprehensive analysis.
- Performance on "Other" Datasets (Table 3): Referring again to Table 3, could the authors explain why the BD (combined set) does not consistently achieve better performance on the "Other" testing datasets compared to its individual subsets?
- Selection of 40 Systems: What was the rationale behind selecting 40 speech generation systems? It would be valuable to understand if all these systems can produce natural-sounding speech.
- Human Verification of Generated Samples: It would also be useful to know if the dataset was subjected to any human verification, as generative models often produce a variety of artefacts, including silences or truncated words. Verifying whether the generated samples are indeed suitable deepfake candidates would add credibility to the dataset.

---

> ### Author Response · Authors · 2024-11-20
> **Response to Official Review of Submission14000 by Reviewer cq2e - Part 1**
>
> We sincerely thank the reviewer for their thorough evaluation of our paper and for providing insightful comments and constructive feedback. Below, we address each of the points raised.
>
> **W1: Multilingual Focus**
>
> We acknowledge the reviewer’s point that English and Chinese are prioritized in our dataset. This is due to practical limitations, as most open-source generation tools primarily support English (and sometimes Chinese). To address this while enabling multilingual research, we structured our dataset into two parts: BD (Bilingual Dataset) for English and Chinese and MD (Multilingual Dataset), which covers 46 languages.
>
> In our experiments, MD is used for cross-lingual evaluation to test whether deepfake detectors can generalize across languages while minimizing the influence of generator differences. The train and dev sets include only English and Chinese data, while the test set is divided into 10 subsets—9 for primary languages and one combined subset for the remaining 37 languages, covering all 46 languages.
>
> In total, MD includes over 1 million samples across 46 languages, with approximately 20,000 samples per language. For our experiments, we selected 5,000 samples per language for testing, leaving the remaining samples as a hidden set. We plan to release this hidden set as well (for both BD and MD) to support further research.
>
> **W2: Coverage of Speech Generation Tools**
>
> Given the numerous existing speech generation methods, many of which overlap or combine variants, it is challenging to cover every method comprehensively. Older generation methods are already well-represented in existing datasets, making it redundant for us to include them again. Instead, we focused on advanced and underrepresented generation methods. We carefully selected better-performing, newer, or representative tools to maximize the diversity and relevance of the dataset.
>
> As speech generation tools continue to evolve, new methods will undoubtedly emerge. While it’s impractical to update the dataset with each new method immediately, we plan to release updated versions periodically once a sufficient number of new methods have accumulated.
>
> **Q1: Generation Process Details**
>
> We have added more details about the generation process in Section 3.1 of the revised paper. The number of utterances generated for each tool is primarily determined by the selected text and corresponding speech from the real datasets, which form the foundation of the generated content. The basic number of utterances is guided by the languages supported by each method. For methods with additional features (e.g., OpenVoice, CosyVoice) or specific use cases (e.g., Tortoise for cross-speaker experiments), we generate supplementary data to explore these aspects. Finally, post-processing steps, such as quality filtering, may result in some utterances being excluded.
>
> **Q2 & Q3: Test on Other Datasets**
>
> We added results on 10 additional test sets in Table 10 for the revised paper, and provide a partial table here. An observation is that models trained on the full BD dataset do not always outperform those trained on its subsets. This may be due to the distinct generation methods in BD-EN and BD-CN, which align more closely with certain test sets. In contrast, the broader diversity of the full BD dataset can sometimes dilute this alignment.
>
> Table 10*: Performance evaluation (EER\%) of model (W2V+AASIST) trained on ASVspoof2019 (ASV19) or Bilingual Dataset (BD) across multiple test sets.
> | Training Dataset | ASV19  | FOR   | WF    | ASV21LA | ASV21DF | CFAD  | ITW   | MLAAD | CD-ADD | ASV24  |
> |------------------|--------|-------|-------|---------|---------|-------|-------|-------|--------|--------|
> | ASV19           | 0.89   | 6.18  | 3.48  | 6.57    | 2.98    | 20.53 | 10.07 | 18.26 | 8.55   | 1.41   |
> | BD              | 2.91   | 6.00  | 0.58  | 7.27    | 2.85    | 12.39 | 2.01  | 12.86 | 2.42   | 0.71   |
> | BD-EN           | 5.28   | 8.33  | 0.96  | 11.91   | 2.97    | 21.42 | 2.62  | 16.75 | 3.54   | 0.71   |
> | BD-CN           | 0.99   | 4.88  | 0.64  | 3.92    | 5.87    | 11.72 | 3.34  | 10.17 | 7.16   | 1.17   |

---

> ### Author Response · Authors · 2024-11-20
> **Response to Official Review of Submission14000 by Reviewer cq2e - Part 2**
>
> **Q4 & Q5: Selection of Generation Systems and Human Verification**
>
> To select speech generation systems, we first identified open-source tools and APIs that claim advanced performance, prioritizing those supported by technical papers or reports. We implemented their released codebases or used their APIs to generate over 100 diverse sample audio clips for each system. These samples underwent human verification to ensure natural-sounding quality. Systems producing artifacts, silences, or other issues were filtered out at this stage. Only after passing this filtering process did we proceed with batch generation to create data covering different languages and voices. After generation, we applied voice activity detection (VAD) to filter out speech segments with less active speech. Additionally, random samples from each system were manually reviewed throughout the process to ensure consistency and naturalness.

---

> > ### Comment · Reviewer_cq2e · 2024-11-27
> > **Review feedback**
> >
> > Thank you for your detailed responses.
> >
> > - While I appreciate the engineering effort, I have some reservations about the long-term utility of this dataset, given the rapid advancements in this field.
> > - Furthermore, without the multilingual component, the approach is a simple extension of VoiceWukong [1] (which uses 34 different synthesis methods for generating 6,800 English and 3,800 Chinese deepfake samples).
> >
> > Additionally, the selection and generation methods seem to lean more towards the engineering aspect rather than introducing fundamentally new methodologies. That said, I still feel that, given the scale, the dataset can be a valuable contribution and will benefit the community.
> > Hence, I would like to retain my score.
> >
> > [1] Yan, Ziwei, Yanjie Zhao, and Haoyu Wang. "VoiceWukong: Benchmarking Deepfake Voice Detection." arXiv preprint arXiv:2409.06348 (2024).

---

> > > ### Author Response · Authors · 2024-11-27
> > > **Response to Reviewer Feedback: Clarifications on Contributions, Utility, and Comparisons**
> > >
> > > Thank you for your thoughtful feedback and acknowledgment of the effort behind our dataset. We deeply value your recognition of its potential contribution to the community and appreciate the opportunity to address your concerns in more detail. We hope the following clarifications provide additional context and insight into our work:
> > >
> > > - **Long-Term Utility**: As noted in our previous response, we recognize the rapid evolution of speech generation tools and the emergence of new methods. While immediate updates with every new method are impractical, we view this work as foundational and plan to release periodic updates as sufficient new techniques accumulate, ensuring the dataset remains relevant and serves as a robust resource for the community.
> > > - **Comparison to VoiceWukong**: We would like to clarify that VoiceWukong and our work are contemporaneous efforts, with VoiceWukong being submitted to arXiv in September 2024.  Given the timing and independent nature of the works, we respectfully suggest that it may not be accurate to describe our work as a simple extension of VoiceWukong. Instead, we see both efforts as complementary contributions to advancing the field.
> > > - **Generation Methods**: We acknowledge that many generation methods in this field overlap, making it difficult to introduce entirely new methodologies. However, compared to prior datasets, our work incorporates the latest cutting-edge generation techniques, which reflect the state-of-the-art in speech synthesis. This ensures that SpeechFake not only captures the current landscape of generation tools but also pushes the boundary by integrating innovative approaches where possible.
> > > - **Concluding Remarks**: In summary, we emphasize that SpeechFake is a large-scale and diverse dataset that integrates cutting-edge techniques and supports multiple languages. We believe this foundational work will provide significant value to the community, and we remain dedicated to periodically updating it to maintain its relevance in the rapidly advancing field of speech generation.
> > >
> > > Thank you once again for your thoughtful review and for considering our clarifications. We humbly hope that these points help to better convey the broader contributions of our work. Your constructive feedback has been invaluable in improving this submission, and we are grateful for the time and effort you have invested in its evaluation.

---

### Official Review · Reviewer_iCvc · 2024-11-05

**Soundness:** 3
**Presentation:** 4
**Contribution:** 2
**Rating:** 6
**Confidence:** 4

**Summary:**

The paper introduces a novel database for speech deepfake detection, comprising a large-scale collection of samples generated by 40 different speech synthesis tools across 46 languages. Benchmark results accompany the dataset, with detailed analyses provided on performance across various subsets, each with a specific focus.

**Strengths:**

The proposed database stands out for its extensive coverage of languages and speech generation tools, providing a robust foundation for deepfake detection research. Additionally, the dataset includes rich metadata, which is invaluable for in-depth analysis of anti-spoofing performance or targeted optimizations. The exploration of generator types and the effects of language and speaker variability is particularly engaging and relevant to recent advances in the field.

**Weaknesses:**

While the dataset is comprehensive, the overall scope is closely aligned with previous work in the field, as acknowledged in Table 1. Although the paper offers a solid contribution, it represents an incremental rather than groundbreaking addition to the anti-spoofing community. The reviewer appreciates the authors' efforts but suggests that more extensive analyses or use cases could further strengthen the contribution. Potential extensions might include exploring additional anti-spoofing model variants, conducting detailed error analyses, or leveraging the dataset for novel applications using pre-trained anti-spoofing models.

**Questions:**

- As noted in the weaknesses, additional analyses would elevate the paper’s contribution. The inclusion of diverse use cases for the dataset could also significantly enhance its impact and utility.
- Why do TTS-Tortoise and VC-OpenVoiceTTS contain significantly more data than other systems?
- Why is BD-UT excluded (or separated) from the result discussion table? Including access dates or version details for the API services would also be advisable, as these may vary over time.

---

> ### Author Response · Authors · 2024-11-20
> **Response to Official Review of Submission14000 by Reviewer iCvc**
>
> We sincerely thank the reviewer for their thorough evaluation of our paper and for providing insightful comments and constructive feedback. Below, we address each of the points raised.
>
> **W1: Dataset Contribution**
>
> We acknowledge that our work is, in some respects, an incremental addition. However, SpeechFake’s large scale, advanced generation techniques, and multilingual coverage address key gaps left by prior datasets and provide essential advancements for the anti-spoofing community. No previous dataset offers this combination of scale and diversity, which is vital for building detection systems that generalize across languages, generation methods, and real-world contexts.
>
> **W2 & Q1:** **Additional Analyses and Use Cases**
>
> We appreciate the suggestion to expand our analyses and use cases. For additional analysis, we prioritized establishing robust baselines and analyzing key factors—generation methods, language diversity, and speaker variation—to align with the dataset’s primary purpose of supporting generalizable and comprehensive evaluation. While further analysis could offer deeper insights, our focus was to provide foundational results that future studies can expand upon. For use cases, we have included several additional use cases in the appendix, such as applications in privacy-preserving deepfake detection, automatic speech recognition robustness, and adversarial attack resilience.
>
> **Q2: More data for TTS-Tortoise and VC-OpenVoiceTTS**
>
> These two systems contain more data because they support a wider variety of voices. For TTS-Tortoise, we generated additional voices within one architecture to study cross-speaker performance. For VC-OpenVoiceTTS, its multi-speaker and style transfer capabilities allowed for more varied outputs, enhancing analysis across voice and style variations in voice conversion.
>
> **Q3: About BD-UT**
>
> We separated BD-UT because it serves as an expansion of our test set, specifically designed as an unseen test set. BD-UT includes high-quality commercial data, and since we lack details about their implementation, it allows us to evaluate whether deepfake detectors can generalize to real-life generation methods effectively. Regarding version details for the APIs, many commercial services do not provide explicit versioning information for their tools. Instead, we include the voice names provided by the APIs in the dataset metadata.

---

> > ### Comment · Reviewer_iCvc · 2024-11-27
> > **Reply to the rebuttal**
> >
> > Thanks for sharing the rebuttal. The rebuttal has generally addressed my concerns. While I do appreciate the great effort in expanding the dataset, the proposed method is more in the pure-empirical side.
> >
> > In general, I would share feelings similar to those of the reviewer cq2e and would keep my score unchanged.

---

> > > ### Author Response · Authors · 2024-11-27
> > > **Response to Reviewer Feedback**
> > >
> > > Thank you for your thoughtful feedback and for sharing your perspective. We appreciate your acknowledgment of the effort behind expanding the dataset and are grateful that our rebuttal has addressed many of your concerns.
> > >
> > > We understand that you share feelings similar to those of reviewer cq2e. To provide additional clarity, we have addressed specific points raised by reviewer cq2e in detail, including clarifications on the foundational nature of our work, the long-term utility of the dataset, and the broader contributions it supports. While these clarifications were part of our response to cq2e, they may also help address shared concerns, and we encourage you to review them if you find them relevant.
> > >
> > > Once again, we thank you for your constructive feedback and for recognizing the value of our work. Your insights have been invaluable, and we remain committed to improving this submission based on the helpful reviews we have received.

---

### Meta-Review · Area_Chair_wAjL · 2024-12-19

**Metareview:**

The paper introduces a large synthetic speech data set, with a particular focus on a wide range of synthesis models and languages.

I recommend a rejection because of a lack of clear demonstrated value, as pointed out by all reviewers.

Novelty of a data set paper should be demonstrated differently compared to a paper that proposes an approach to solving a problem. This paper did well demonstrating the wide variety of synthesis models and languages, but fails to show the value (i.e., challenge) that the data set brings. The problem is a combination of model choice and writing. The paper should really stress test the best models out there to demonstrate a clear challenge. Reviewers have given valuable suggestions, and the authors are encouraged to improve the paper based on the feedback.

The authors are also encouraged to improve the reproducibility of the data set so that others are able to automate/reproduce the process in future settings.

**Additional Comments On Reviewer Discussion:**

The authors had done a great job resolving clarification issues. However, the fundamental problem of around the utility of the data set was still present.

---

### Decision · Program_Chairs · 2025-01-22

Reject